# Antibiotic Therapy for Pulmonary Exacerbations in Cystic Fibrosis—A Single-Centre Prospective Observational Study

**DOI:** 10.3390/antibiotics12040734

**Published:** 2023-04-09

**Authors:** Carsten Schwarz, Eliana Wimmer, Frederik Holz, Claudia Grehn, Doris Staab, Patience Ndidi Eschenhagen

**Affiliations:** 1Cystic Fibrosis Center, Clinic Westbrandenburg, 14467 Potsdam, Germany; 2Department of Education and Research, HMU Health and Medical University Potsdam, 14471 Potsdam, Germany; 3Paediatric Practice at Traveplatz, Dr. Kilger, Dr. Kabelitz, Dr. Shetty, 10247 Berlin, Germany; 4Department of Pediatric Respiratory Medicine, Immunology and Critical Care Medicine, Charité–Universitätsmedizin Berlin, Corporate Member of Freie Universität Berlin, Humboldt-Universität zu Berlin, and Berlin Institute of Health, 13353 Berlin, Germany; 5BIH Berlin Institute of Health, Charité University Medicine Berlin, 13353 Berlin, Germany

**Keywords:** cystic fibrosis, bronchopulmonary exacerbation, C-reactive protein, home spirometry

## Abstract

People with cystic fibrosis experience bronchopulmonary exacerbations, leading to lung damage, lung function decline, increased mortality, and a poor health-related quality of life. To date, there are still open questions regarding the rationale for antibiotic use and the optimal duration of antibiotic therapy. This prospective single-center study (DRKS00012924) analyzes exacerbation treatment over 28 days in 96 pediatric and adult people with cystic fibrosis who started oral and/or intravenous antibiotic therapy in an inpatient or outpatient setting after clinician diagnosis of bronchopulmonary exacerbation. Biomarkers of exacerbation were examined in terms of their ability to predict response to treatment and the need for antibiotic therapy. The mean duration of antibiotic therapy was 14 days. Inpatient treatment was associated with a poorer health status, but no significant difference was found in the modified Fuchs exacerbation score between inpatients and outpatients. A significant increase of in-hospital FEV1, home spirometry FEV1, and body-mass index and a significant decrease of the modified Fuchs symptom score, C-reactive protein, and 8 out of the 12 domain scores of the revised cystic fibrosis questionnaire were demonstrated after 28 days. However, a trend towards a FEV1 decline in the inpatient group on day 28 could be demonstrated, while FEV1 was maintained in the outpatient group. Correlation analyses of changes between baseline and day 28 show a strong positive correlation between home spirometry and in-hospital FEV1, strong negative correlations between FEV1 and the modified Fuchs exacerbation score and between FEV1 and C-reactive protein, and a moderately negative correlation between FEV1 and the three domains of the revised cystic fibrosis questionnaire. Responders and non-responders to antibiotic therapy were defined in terms of FEV1 improvement after therapy. A higher baseline C-reactive protein, a greater decrease in C-reactive protein, a higher baseline modified Fuchs exacerbation score, and a greater decrease in the score after 28 days could be found in the responder group, while other baseline and follow-up parameters like FEV1 showed no significant differences. Our data show that the modified Fuchs exacerbation score is applicable in a clinical setting and can detect acute exacerbations regardless of health status. Home spirometry is a useful tool for outpatient exacerbation management. A change in C-reactive protein and a modified Fuchs score change are suitable follow-up markers of exacerbation due to their strong correlation with FEV1. Further studies are needed to assess which patients would benefit from a longer duration of antibiotic therapy. C-reactive protein at exacerbation onset and C-reactive protein decline during and after therapy better predict antibiotic therapy success than FEV1 at therapy onset, while the modified Fuchs score indicates exacerbation regardless of the need for antibiotic therapy, suggesting that antibiotic therapy is only part of exacerbation management.

## 1. Introduction

Cystic fibrosis (CF) is one of the most common monogenetic, autosomal recessive inherited diseases. The estimated incidence ranges from 1:1353 in Ireland to 1:128.434 in China; however, it is still not known in many parts of the world [1,2,3]. Loss-of-function mutations in the cystic fibrosis transmembrane conductance regulator (CFTR) gene encoding for the CFTR channel cause a reduction in chloride and bicarbonate ion transport at cell membranes in many organs. This results in a multi-organ disease in the lung, leading to chronic inflammation and infection with respiratory pathogens and recurrent episodes of worsening symptoms called bronchopulmonary exacerbations (BEx). Progressive destruction of lung tissue with respiratory failure is the most common cause of death in CF, and BEx account for most hospital admissions in people with cystic fibrosis (pwCF) [4]. Advances in therapy have led to dramatically increased life expectancy and quality of life, though a substantial number of patients are still severely affected [4]. The symptomatic therapy of CF lung disease includes mucoactive agents, inhalative and systemic antibiotics, bronchodilators, and anti-inflammatory drugs. During the last decade, highly effective CFTR modulators (HEMT) have been developed and approved for many CFTR mutations. HEMT provides a causal treatment, improving CFTR ion transport [5] or acting as molecular chaperones [6,7]. Today, about 90% of pwCF in the European and North American CF populations are eligible for modulator therapy [8]. HEMT reduces the rate of severe BEx [9], but long-term data show that a significant number of pwCF continue to be affected, particularly by milder BEx [10]. BEx drive disease progression, increase morbidity and mortality, and worsen health-related quality of life in pwCF [4]. The pathophysiology of BEx results from a complex interaction between host and bacterial pathogen and is not yet fully understood [11]. The presence of other pathogens, like respiratory viruses, often seems to play a role in the initiation of BEx [12]. However, chronic fungal colonization is also associated with more frequent BEx [13,14,15]. Most BExs are due to the clonal expansion of preexisting bacterial strains, but they may also be caused by the acquisition of a new bacterial pathogen [11]. Due to overlapping symptoms, it may be difficult to identify and treat the most relevant respiratory pathogen at a given time. In addition, pathogens may be alternately relevant during one BEx course, thus an initially predominantly viral infection may develop into a bacterial and eventually fungal infection [11,13]. Inhalative antiseptic therapies that treat all respiratory pathogens simultaneously and particularly the most proliferating pathogen could therefore prove useful for exacerbation therapy in addition to other antimicrobial therapies [14,15].

To standardize the BEx definition for clinical trials, in 2011, the EuroCareCF group modified the preexisting Fuchs BEx score [16] and defined BEx as a recent change of at least two out of a list of six items requiring additional antibiotic therapy [17]: I. change in sputum volume or color; II. increased cough; III. increased malaise, fatigue, or lethargy; IV. anorexia or weight loss; V. decrease in pulmonary function by 10% or more and/or radiographic changes; and VI. increased dyspnea. Several studies have now demonstrated that the Fuchs-BEx score inadequately reflects the clinician’s diagnosis of BEx. For example, in a phase III clinical trial with 751 respiratory events, more than one third had ≥4 Fuchs criteria present but failed to be assessed as BEx, and only 6/12 Fuchs criteria were present more often in the assumed BEx group than in the non-BEx group [18]. However, it is also under discussion whether the modified Fuchs BEx score is truly more applicable to the real world [19,20]. Recently, a decline in FEV1 (ΔFEV1) as modest as 5% or more could be associated with increased cough and/or sputum and clinician-diagnosed BEx in children with CF [21], questioning the requirement for ΔFEV1 ≥10%. This is particularly important in the HEMT era with lower sputum volumes and potentially milder FEV1 changes, where ΔFEV1 ≥10% may no longer prove sensitive enough. The search for biomarkers that clearly indicate the need for antibiotics in BEx remains unsolved, and it may be doubted whether ΔFEV1 is such a marker at all. In a recent study, BEx groups could be defined in relation to viral or bacterial involvement, respectively, and systemic inflammation, confirming a previous study that found the highest ΔFEV1 in mixed viral-bacterial BEx and not in only bacterial-induced BEx [22,23]. Moreover, BEx without FEV1 changes may exist [24]. Other important issues are the monitoring of treatment response and the duration of antibiotic therapy. The only yet existing randomized trial specifically examining treatment duration for BEx in pwCF, the STOP2 (Standard Treatment of Pulmonary exacerbation 2) trial, identified a duration of 10 vs. 14 days for early responders and 14 vs. 21 days for non-early responders as equivalent in terms of FEV1 outcome, and other studies achieved similar results [25,26]. However, in the STOP2 trial, a trend in the shorter treatment groups towards a worse FEV1 outcome after completion of therapy is evident, as is a trend towards a shorter time to the next BEx in both groups with shorter treatment. These trends are not statistically significant but may cumulate to a worse long-term pulmonary outcome after multiple BEx, so that further randomized studies with a longer follow-up period are still desirable. Many parameters, especially C-reactive protein (CrP) and FEV1, have been extensively studied for their utility in assessing response to therapy, but no association between these parameters could be found [27]. In the STOP trial, the change in FEV1 and the CRISS Score, a patient-reported symptom diary, were found to be positively correlated [28,29]. A change in several domains of the CF-specific patient-reported outcome measure was reported. The CFQ-R (cystic fibrosis questionnaire-revised) has already been found to moderately correlate with BEx [30] and to change similarly to ppFEV1, but statistical analysis has not been performed [31]. A correlation between ΔFEV1 and the modified Fuchs BEx symptom score during BEx treatment has not yet been examined outside clinical trials. During a randomized clinical trial, vast differences between investigator and modified Fuchs score BEx definition were found [18]. In summary, there are still open questions and a lack of evidence about the definition and diagnosis of BEx, the timing of treatment initiation, the duration of treatment, and how treatment should be monitored. The management of pulmonary exacerbations is a crucial factor influencing the outcome of pwCF. BEx worsens long-term health status, but inpatient treatment is also associated with a poorer quality of life [32]. Additionally, rational prescribing of antibiotic therapies in CF is particularly challenging. It is therefore important to identify biomarkers that indicate the need for BEx treatment and response to therapy and that may identify the need for antibiotic therapy. In addition, it is important to establish objective outcome parameters that can be measured at home to enable safe and effective outpatient BEx therapy. We therefore aimed to analyze BEx treatment and outcome parameters in pwCF in our center to identify biomarkers suitable for BEx management in the clinical routine.

## 2. Results

### 2.1. Study Cohort

A total of 105 patients were included in the study. Due to missing data, nine patients were excluded, and statistical analyses were finally performed for 96 patients with complete home spirometry data. For the analysis of further outcome parameters and the analysis of response to therapy, only patients with complete data on day 0 and day 28 were included in the statistical analysis (see Figure 1).

### 2.2. Baseline Data/Demographical Data, BEx Treatment, and Sputum Microbiology

Statistical analysis of the baseline and demographic data revealed significant differences between inpatient and outpatient groups (please see Table 1). Inpatient treatment was associated with a poorer health status, as indicated by a lower median body mass index (BMI), a higher proportion of pancreatic insufficiency, a higher rate of *Pseudomonas aeruginosa* and fungi detection in Day 0 sputum samples, a more rapid FEV1 (forced expiratory volume in one second) decline during the past two years, a lower baseline FEV1, and a higher hospitalization rate in the two years prior to study entry. Fitting this, the inpatient group had a significantly worse health-related quality of life (HRQL) assessed by CFQ-R questionnaires, especially for the domains “physical functioning”, “body image”, and “respiratory symptoms.” Inpatients also had a higher median leukocyte count at study entry, indicating a more severe exacerbation. However, the modified Fuchs score demonstrated no significant difference between the two groups, which shows that the score is able to identify acute exacerbations regardless of disease severity.

In Table 2, BEx treatment is presented, with 100% of inpatients receiving intravenous antibiotic therapy and a vast majority of outpatients receiving oral antibiotic therapy. Inpatients were more often treated with additional steroids and antifungals as part of the BEx therapy. This again points out the more severe phenotype but may also be caused by the fact that inpatients are available for more differentiated treatment decisions. As all therapeutic decisions were made by clinicians regardless of inclusion in the study, these data reflect the typical routine clinical practice of a single center.

### 2.3. Outcome Data

Table 3 and Table 4 give an overview of the parameters collected on days 0 and 28 and their changes, as well as the differences between inpatients and outpatients on day 28. Only cases with complete parameters on days 0 and 28 were analyzed. 

In mean, patients showed response to BEx treatment, as evidenced by significantly increased in-hospital ppFEV1, home spirometry FEV1, and BM, and significantly decreased modified Fuchs BEx symptom score, CrP, and 8 out of 12 CFQ-R domain scores. No significant change was seen in the leukocyte count between days 0 and 28. However, this may be due to the higher proportion of steroid therapy in the inpatient group. In the following, changes in home spirometry FEV1, CrP, the modified Fuchs symptom score, and CFQ-R during the study period are analyzed and compared to the change in in-hospital FEV1 as the gold standard in terms of their suitability as BEx outcome markers. According to their FEV1 change during the study period, patients were classified as responders and non-responders, and the markers collected were assessed as markers of response to BEx therapy.

#### 2.3.1. Pulmonary Function Tests 

Home spirometry showed a significant mean FEV1 increase between days 0 and 28, with the strongest increase already on day 7 in the outpatient group and not before day 14 in the inpatient group, indicating a later response to therapy in this group (please see Figure 2). In-hospital ppFEV1 similarly increased significantly from day 0 to day 28. In both groups, this increase was maintained during the follow-up period. In the outpatient group, a further significant increase is recorded between days 21 and 28, while in the inpatient group, there is a trend towards a decline in FEV1 after the mean end of therapy on day 14. This again underlines the poorer health status of the inpatient group and may indicate that a longer antibiotic treatment could be beneficial for at least a part of this group. The second FEV1 increase after 28 days in the outpatient group may be due to other influencing factors, e.g., resolving viral infections.

Bivariate correlation analysis showed a strong positive correlation between home spirometry and in-hospital FEV1 (please see Figure 3), suggesting that home spirometry may be a valuable tool in assessing the need for and response to BEx therapy in CF in an outpatient setting.

#### 2.3.2. Modified Fuchs BEx Score

The change in the modified Fuchs BEx score over the study period is displayed in Figure 4A,B. The majority of patients achieved a score of 0 or 1 after 28 days, indicating response to BEx therapy (see Figure 4A). The corresponding FEV1 increase is shown in Figure 4B. Similarly, correlation analysis between ΔppFEV1 and the change in the modified Fuchs BEx score on days 0 and 28 shows a strong negative correlation (please see Figure 5). On day 28, the score again differed significantly between outpatients and inpatients, fitting the trend towards FEV1 decline on day 28 in the outpatient group. These results suggest that the BEx score, similar to FEV1, is also a suitable marker of response to BEx therapy in an outpatient setting. 

#### 2.3.3. C-Reactive Protein 

CRP also changed significantly on day 28 (please see Figure 6). The correlation analysis between change in CRP and ΔFEV1 also showed a strong negative correlation without a significant difference between inpatient and outpatient groups (please see Figure 7), suggesting that change in CRP is a valuable surrogate parameter for response to BEx therapy.

#### 2.3.4. Correlation Analysis of CFQ-R and ΔFEV1

Correlations of medium effect strength were found between ΔFEV1 and some domains of the CFQ-R, with the strongest correlations shown for the domains “energy”, “physical functioning”, and “respiratory symptoms” (see Figure 8).

#### 2.3.5. Assessment of Response to Therapy

To assess factors influencing response to therapy, we defined a home spirometry FEV1 increase of at least 5% between day 0 and day 28 as successful BEx treatment and a FEV1 increase <5% as a poor response to therapy. According to this definition, 57 patients were classified as treatment responders, while 39 patients were classified as non-responders (please see Table 5). A comparison of responders and non-responders in terms of baseline and outcome parameters showed no significant differences in age, sex, BMI, pancreatic insufficiency, diabetes, airway colonization, leukocyte count, hospitalization rate, FEV1 at baseline and FEV1 decline in the two years prior to the study, duration of therapy, or inpatient vs. outpatient treatment. A significant difference between responders and non-responders was seen in the mean baseline CrP at day 0, which was higher in the responder group. In addition, responders showed a significantly greater decrease in CrP from day 0 to day 28. The responder group had significantly higher BEx symptom scores on day 0 and a greater decrease in BEx symptom scores between days 0 and 28. The fact that the non-responders started with better baseline values but had a worse therapy outcome could be due to BEx being influenced by factors that do not respond to antibiotic treatment, such as viral infections or inflammatory processes, for which other treatment options should be explored. This is also supported by the correlation between the success of antibiotic therapy and CRP. Thus, CRP is the best predictor of the need for antibiotic therapy for BEx in pwCF, and the modified Fuchs BEx score seems to best reflect the state of BEx, mostly independent of the need for antibiotic therapy.

## 3. Discussion

In our study, FEV1, an established marker in clinical trials, was used as a reference parameter to examine biomarkers for exacerbation outcome. A strong correlation between home and in-hospital spirometry was demonstrated, confirming other studies [33] showing that home spirometry is a suitable BEx treatment monitoring tool in CF. FEV1 increased quickly and significantly in the first seven days. However, outpatients showed an early FEV1 increase on day 7, but the mean greatest change in inpatients happened not before day 14, implying a higher number of late therapy responders in the inpatient group and a possible need for longer antibiotic therapy duration. These results are in line with the findings of the only yet available randomized trial assessing therapy duration in BEx in CF (STOP2). It determined a treatment duration of 7–10 days for early responders and of 14–21 days for late responders [25]. Of note, this differs from a systematic review examining 52 studies in which no benefit from a therapy duration longer than 10–12 days was found [26]. In our study, no significant differences in treatment duration existed between inpatients and outpatients, which is most likely explained by the fact that therapy duration was determined at the beginning and was not dependent on the clinical outcome, especially in outpatients.

In our cohort, the FEV1 increase is not equally maintained, as on day 28, the inpatient group shows a decrease and the outpatient group even shows an increase. The decrease at day 28 implies that a subgroup of inpatients may need a longer treatment duration to maintain therapy success, as discussed above. The FEV1 increase on day 28 in outpatients indicates that there may be other factors other than antibiotic treatment influencing BEx outcome, e.g., viral infections or inflammatory processes. These factors have already been addressed [11], and it may be beneficial to systematically investigate them to better predict the need for and duration of antibiotic therapy.

The modified Fuchs BEx score did not differ significantly between inpatients and outpatients, despite the poorer health status of the inpatients. It has therefore been proven that the score adequately reflects the actual changes and the need for BEx management, mostly independent of the overall health status. In addition, we were able to demonstrate, in contrast to other studies [18], a strong correlation between the improvement of the modified Fuchs BEx score and mean ΔFEV1 after therapy, which makes it a suitable tool for BEx therapy monitoring in a real-world setting for inpatients and outpatients.

CRP was demonstrated to be a valuable surrogate parameter for response to BEx therapy. We could also demonstrate that CRP and FEV1 strongly correlate, and that CRP level at onset of BEx and CRP decline during treatment better predict the success of antibiotic therapy than FEV1.

This is in line with other studies demonstrating that ΔFEV1 insufficiently reflects the need for antibiotics in BEx [23]. CRP therefore proved to be the best predictor of BEx with need for antibiotic treatment in our study. In comparison, the modified Fuchs BEx score seems to best reflect the state of BEx, mostly independent of the need for antibiotic therapy. Furthermore, this indicates that factors other than bacterial proliferation additionally play an important role in BEx, such as viral and fungal infections and inflammatory processes triggered by complex heterogeneous immune responses to antigens in the airways of pwCF [34,35,36]. In a recent study, CRP worked successfully in combination with a respiratory virus panel to classify BEx subphenotypes [22].

## 4. Materials and Methods

### 4.1. Study Design and Patient Selection

This nonrandomized, prospective single-center observational trial (DRKS00012924) was conducted over 10 months, from September 2016 to July 2017. PwCF with acute BEx who were started on antibiotic therapy either as outpatients or inpatients were recruited for the study on the day of their presentation (day 0; see Figure 9). Therapeutic decisions prior to and during the study were made by clinicians regardless of inclusion in the study. The selection criteria for including pwCF were as follows:No history of lung transplantation;≥6 years or ≤75 years;acute BEx (defined by ≥2/6 positive items in the modified Fuchs score published by Bilton et al. [17]);ability to perform lung function and home spirometry;being capable of giving consent;subject (or legal guardian) has given written consent to participate in the study.

Exclusion criteria:
History of lung transplantation;<6 years or >75 years;no acute BEx (defined by ≥2/6 positive items in the modified Fuchs score published by Bilton et al. [17]);non-ability to perform lung function and home spirometry;being capable of giving consent;subject (or legal guardian) has not given written consent to participate in the study.

After recruitment, symptoms of BEx were standardized using a questionnaire proposed by Fuchs et al. in 1994 and modified by Bilton et al. in 2011 for the EuroCF Group [16,17], which contained the following six items:Change in sputum volume or color;increased cough;increased malaise, fatigue, or lethargy;anorexia or weight loss;decrease in FEV1 by 10% or more/radiographic changes;increased dyspnea.

CFQ-R, in-hospital lung function, C-reactive protein, sputum microbiology, and body mass index were the additional parameters that were optionally collected. Follow-up examinations with home spirometry and collection of the symptom questionnaire were carried out at home on days 7, 14, and 21. All parameters were collected again during a follow-up examination on site on day 28 (see Figure 9). For home spirometry, patients were given a respiratory monitor (asma-1TM, model 4000, BS EN ISO 23747: 2007, manufacturer: Vitalograph^®^) on day 0. The handling of the device was explained in detail to each patient to ensure that they were able to perform the measurements independently. The measurement consisted of a series of three FEV1 measurements taken a few minutes apart. The CFQ-R questionnaire has nine quality of life domains (physical, role/school, vitality, emotion, social, body image, eating, treatment burden, and health perceptions) and three symptom scales (weight, respiratory, and digestion), each containing several items. The items are summed to produce range scores between 0 and 100, with higher values representing better health. Three different versions of the CFQ-R questionnaire were used depending on age group [37,38].

### 4.2. Data Collection and Statistical Analysis

In addition to the outcome parameters, baseline data were collected using patient records and the German patient registry software “Muko.web”. All patients with correctly performed home spirometry and a complete symptom score on each study day (day 0, day 7, day 14, day 21, and day 28) were included in the final analysis. For the analysis of the secondary outcome parameter, several subgroups with smaller numbers of patients were formed depending on the number of patients with documented optional data.

The distribution of data was assessed with the Kolmogorov-Smirnov Z-test and the Shapiro-Wilk test for normal distribution. Due to statistical requirements, *t*-tests, Mann-Whitney U-tests, χ2-tests, Fisher exact tests, and Wilcoxon tests were considered suitable for the comparison between two groups. Correlation analyses were conducted to determine the correlation quotient either according to Pearson or according to Spearman, and a confidence interval was calculated for both correlation coefficients using Fisher’s z-transformation. Effect sizes were calculated (e.g., effect size according to Cohen (d) and correlation coefficient according to Pearson (r)). Cohen’s classification was used to assess the effect size.

Data analyses were performed using “SPSS for Windows” version 24.0, Microsoft Excel 2016, OpenOffice Calc version 4.1.1, and R version 4.2.1. The Institute for Biometry and Clinical Epidemiology at Charité, University of Medicine, Berlin (Prof. Dr. Geraldine Rauch) was consulted for statistical advice.

## 5. Conclusions

In summary, our data show that a poorer health status in CF is associated with intravenous and inpatient treatment for BEx. In addition, we were able to identify biomarkers associated with BEx and the need for antibiotics that are useful for clinical routine and home monitoring, including the modified Fuchs BEx score, which could be confirmed in a clinical setting and is able to detect acute BEx regardless of health status. Home spirometry FEV1 change, CRP change, and modified Fuchs score change are useful tools in BEx management for inpatients and outpatients due to strong correlations with in-hospital FEV1 change. Further studies are needed to assess which patients would benefit from a longer duration of antibiotic therapy. CRP at BEx onset and CRP decline best predict BEx with need for antibiotic therapy, while the modified Fuchs BEx score reflects BEx independent of the need for antibiotics. This indicates that antibiotic therapy is only part of BEx management. Therefore, future efforts should further focus on identifying and targeting all BEx-influencing factors, such as viral and fungal infections and inflammatory processes, to enable rational prescribing of antibiotic therapy in pwCF.

## Figures and Tables

**Figure 1 antibiotics-12-00734-f001:**
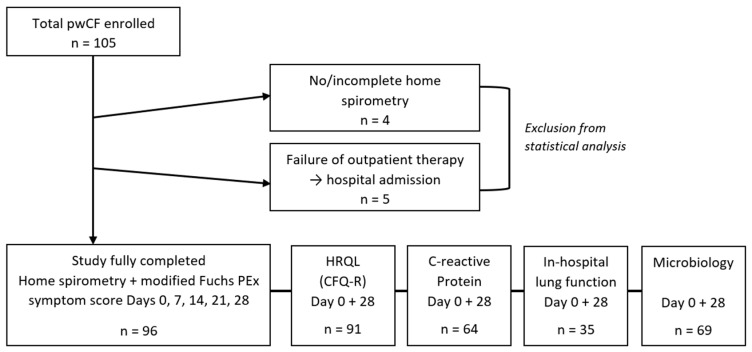
Study cohort.

**Figure 2 antibiotics-12-00734-f002:**
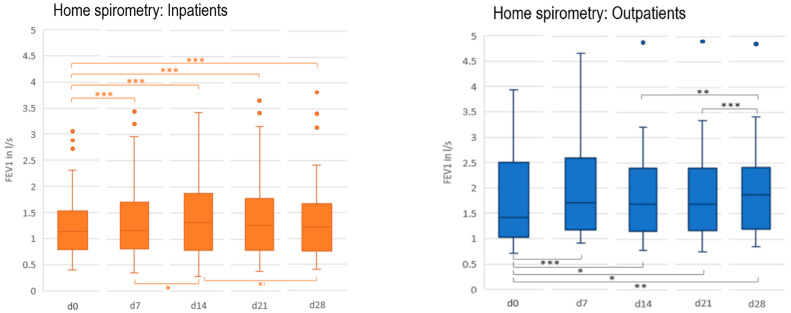
Home spirometry FEV1 (l/s) inpatients and outpatients, days 0–28. * *p* < 0.05, ** *p* < 0.01, *** *p* < 0.001.

**Figure 3 antibiotics-12-00734-f003:**
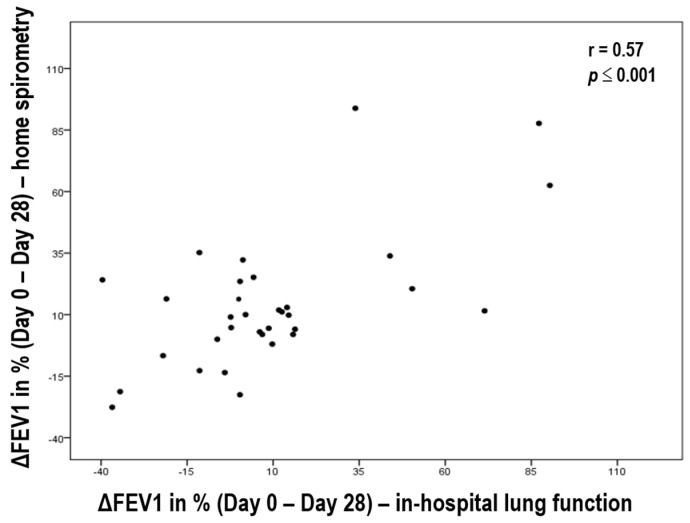
Correlation between home spirometry ΔFEV1 and in-hospital ΔFEV1 days 0–28.

**Figure 4 antibiotics-12-00734-f004:**
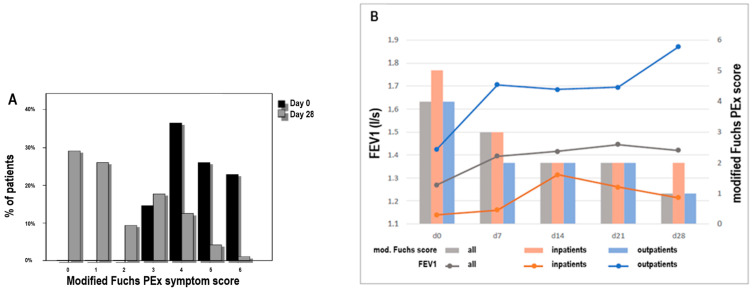
(**A**) Modified Fuchs BEx symptom score day 0 + day 28. Change between Day 0 and Day 28, % of patients. (**B**) Home spirometry + modified Fuchs BEx symptom score, day 0–28. The mean home spirometry FEV1 and Fuchs score from the start of therapy (day 0) and over the whole follow-up period (days 7, 14, 21, and 28) is presented for all patients (grey) and separately for inpatients (orange) and outpatients (blue).

**Figure 5 antibiotics-12-00734-f005:**
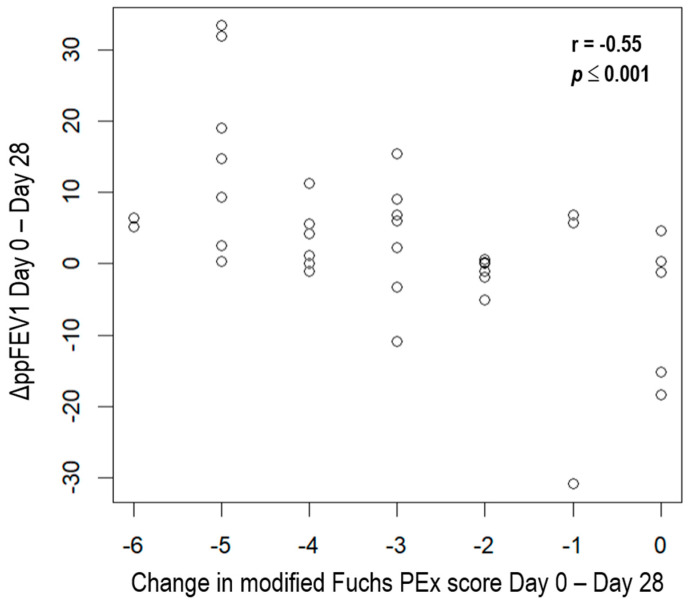
Correlation analysis ΔppFEV1/modified Fuchs score (days 0–28).

**Figure 6 antibiotics-12-00734-f006:**
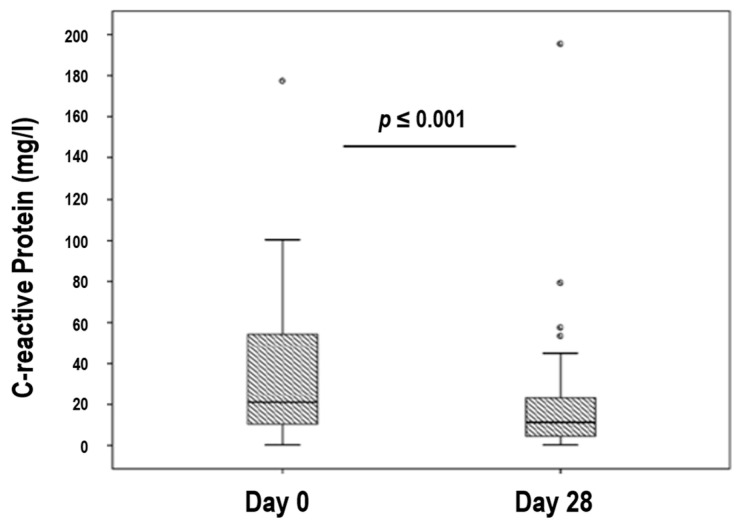
Change in C-reactive protein, days 0 and 28.

**Figure 7 antibiotics-12-00734-f007:**
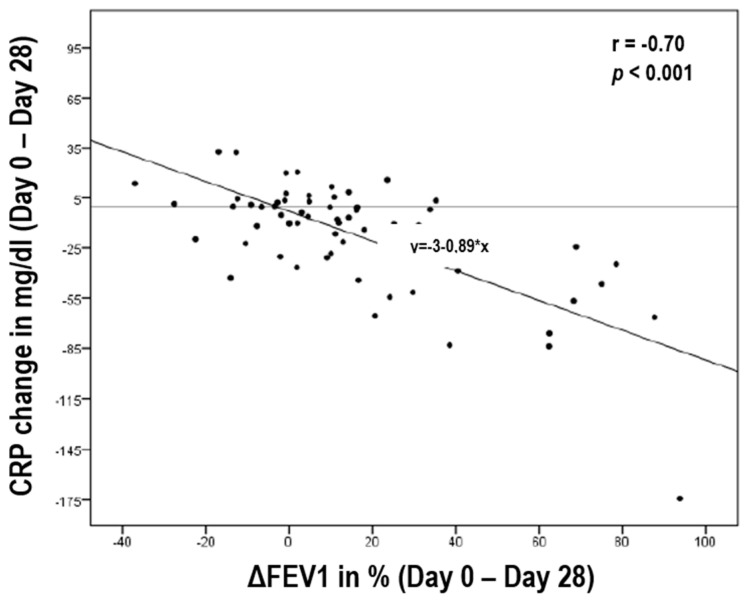
Correlation analysis of ΔFEV1/CRP change (days 0–28), y = regression line.

**Figure 8 antibiotics-12-00734-f008:**
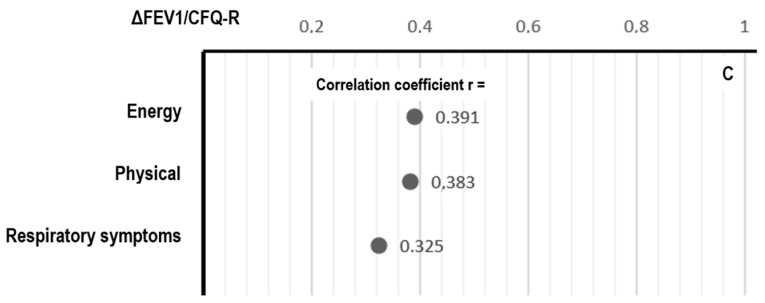
Correlation analysis of ΔFEV1/CFQ-R (days 0–28).

**Figure 9 antibiotics-12-00734-f009:**
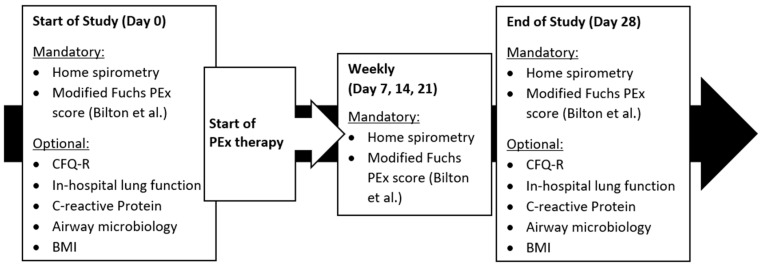
Study design.

**Table 1 antibiotics-12-00734-t001:** Baseline Data (Day 0).

Baseline Data (Day 0) ^a^	All Subjects (*n* = 96)	Outpatients (*n* = 42)	Inpatients (*n* = 54)	*p*-Value ^b^
Gender: female/male, number	50/46 (52.1/47.9)	22 (52.4)	28 (51.9)	ns ^c^
Age	28.5 ± 11.5 (8.8–57.9)	29.2 ± 11.4 (17.5–57.9)	26.95 ± 11.53 (8.8–57.0)	ns
Age < 18 years	10 (10.4)	1 (2.4)	9 (16.7)	*p* ≤ 0.05
BMI (kg/m²)	19.9 ± 3.7 (12.2–33.7)	20.4 ± 3.9 (14.3–33.7)	19.15 ± 3.11 (12.2–28.8)	*p* ≤ 0.001
CF- related diabetes mellitus	38 (39.6)	15 (35.7)	23 (42.6)	ns
Exocrine pancreatic insufficiency	89 (92.7)	36 (85.7)	53 (98.1)	*p* ≤ 0.05
Chronic airway infection: Pseudomonas aeruginosa Staphylococcus aureus	60 (62.5) 41 (42.7)	23 (54.8) 25 (59.5)	37 (68.5) 16 (29.6)	ns *p* ≤ 0.01
Hospitalizations last 2 years (number)	3 ± 4 (0–17)	2 ± 3 (0–11)	4 ± 4 (0–17)	*p* ≤ 0.01
PEx last 2 years (number)	4 ± 4 (0–18)	4 ± 3 (0–14)	4 ± 4 (0–18)	ns
ΔFEV1 last 2 years: ΔFEV1 in %	(*n* = 82) −6.9 ± 14.7 (−58.5–28.9)	(*n* = 33) −4.1 ± 11.5 (−21.9–28.9)	(*n* = 49) −8.7 ± 15.1 (−58.5–13.9)	*p* ≤ 0.01
Home spirometry FEV1 (l/s)	1.3 ± 0.8 (0.4–3.2)	1.4 ± 0.8 (0.7–3.9)	1.1 ± 0.6 (0.4–3.1)	*p* ≤ 0.001
Modified Fuchs PEx symptom-score	4 ± 1 (3–6)	4 ± 1 (3–6)	5 ± 1 (3–6)	ns
Lung function (spirometry): FEV1 (L/s) FVC (L) MEF 25 (L/s) MEF 25/75 (L/s)	(*n* = 59) 1.4 ± 0.9 (0.5–3.7) 2.6 ± 1.0 (1.1–5.7) 0.3 ± 0.3 (0.1–1.4) 0.7 ± 0.8 (0.2–3.3)	(*n* = 34) 1.9 ± 0.9 (0.7–3.7) 2.9 ± 1.0 (1.5–5.6) 0.3 ± 0.4 (0.1–1.4) 0.8 ± 1.0 (0.3–3.3)	(*n* = 25) 1.2 ± 0.6 (0.5 ± 3.0) 2.2 ± 0.8 (1.1–4.1) 0.2 ± 0.3 (0.1–1.3) 0.6 ± 0.6 (0.2–2.4)	*p* ≤ 0.001 *p* ≤ 0.001 *p* ≤ 0.01 *p* ≤ 0.01
Laboratory results: CRP (mg/L) Leukocytes (/nL)	(*n* = 83) 20.3 ± 31.7 (0.3–177.5) 13.1 ± 5.1 (4.1–29.3)	(*n* = 29) 15.4 ± 21.5 (0.3–72.0) 11.0 ± 3.9 (5.2–18.8)	(*n* = 54) 21.1 ± 35.3 (0.8–177.5) 13.6 ± 5.2 (4.1–29.3)	ns *p* ≤ 0.01
CFQ-R domains (sum of item scores) Physical functioning Vitality Emotional functioning Social role/everyday life Body image Eating disturbances Treatment burden Health perceptions Weight Respiratory symptoms Digestive symptoms	(*n* = 91) 42 ± 25 (0–92) 33 ± 18 (0–75) 67 ± 18 (17–100) 56 ± 18 (22–100) 58 ± 26 (0–100) 67 ± 25 (22–100) 78 ± 24 (11–100) 56 ± 18 (11–100) 33 ± 22 (0–100) 67 ± 38 (0–100) 39 ± 20 (0–78) 89 ± 20 (22–100)	(*n* = 40) 56 ± 24 (17–92) 38 ± 18 (0–75) 73 ± 17 (20–100) 61 ± 21 (22–100) 67 ± 24 (17–100) 67 ± 24 (22–100) 89 ± 22 (11–100) 56 ± 18 (11–100) 39 ± 25 (0–100) 44 ± 19 (11–78) 89 ± 15 (44–100) 100 ± 36 (0–100)	(*n* = 51) 33 ± 21 (0–67) 33 ± 18 (0–75) 60 ± 19 (17–93) 50 ± 15 (22–100) 56 ± 26 (0–100) 44 ± 24 (22–100) 78 ± 25 (11–100) 56 ± 18 (11–89) 33 ± 20 (0–78) 28 ± 19 (0–72) 78 ± 22 (22–100) 67 ± 39 (0–100)	*p* ≤ 0.001 ns *p* ≤ 0.05 ns ns *p* ≤ 0.001 *p* ≤ 0.05 ns *p* ≤ 0.05 *p* ≤ 0.05 *p* ≤ 0.01 ns
Microbiology: *Pseudomonas aeruginosa* *Staphylococcus aureus* Fungi	(*n* = 93) 57 (61.3) 44 (47.3) 73 (78.5)	(*n* = 39) 21 (53.9) 27 (69.2) 26 (66.7)	(*n* = 54) 36 (66.7) 17 (31.5) 47 (87.0)	ns *p* ≤ 0.001 *p* ≤ 0.05

^a^ Data are displayed in the form of the median ± SD (min–max) or number (%). ^b^
*p*-values refer to significant differences between the “outpatient” and “inpatient” groups. ^c^ ns = non-significant.

**Table 2 antibiotics-12-00734-t002:** BEx treatment. Type of therapy, total numbers, and percentages of inpatients and outpatients.

Exacerbation Therapy	All Subjects (*n* = 96)	Outpatients (*n* =42)	Inpatients (*n* = 54)	*p*-Value ^b^
Duration of therapy (days) Combined antibiotic therapy Intravenous antibiotic therapy Oral antibiotic therapy Inhalative antibiotic therapy Corticosteroid therapy Antifungal therapy	14 ± 5 (10–28) 62 (64.6) 54 (56.3) 43 (44.8) 20 (20.8) 38 (39.6) 26 (27.1)	14 ± 7 (10–28) 9 (21.4) 0 (0.0) 39 (92.9) 12 (28.6) 6 (14.3) 0 (0.0)	28 (51.9) 53 (98.1) 54 (100.0) 5 (9.3) 8 (14.8) 32 (59.3) 26 (48.1)	ns ^c^ *p* < 0.001 *p* < 0.001 *p* < 0.001 ns *p* < 0.001 *p* < 0.001

Data are displayed in the form of the median ± SD (min–max) or number (%). ^b^
*p*-values refer to significances between outpatients and inpatients. ^c^ ns = not significant.

**Table 3 antibiotics-12-00734-t003:** Outcome days 0 and 28. Analysis shown for patients with complete data on days 0 and 28.

Outcome Day 0–28	Day0 Median ± SD (Min–Max)	Day28 Median ± SD (Min–Max)	*p*-Value
Home spirometry FEV1 (L/s) ^a^	1.3 ± 0.8 (0.4–3.2)	1.4 ± 0.6 (0.4–4.8)	*p* ≤ 0.001
PEx symptom score ^a^	4 ± 1 (3–6)	1± 2 (0–6)	*p* ≤ 0.001
Laboratory parameters ^b^: CRP (mg/L) Leukocyte count (/nL)	21.1 ± 33.1 (0.3–177.5) 12.7 ± 5.1 (4.1–29.3)	11.5 ± 27.5 (0.3–195.6) 11.9 ± 5.4 (2.4–25.5)	*p* ≤ 0.001 ns
In-hospital lung function ^c^: ppFEV1	43 ± 20.42 (16–101)	52.5 ± 21 (17–107)	*p* ≤ 0.001
CFQ-R domains (%) ^d^ PHYSICAL FUNCTIONING VITALITY EMOTIONAL FUNCTIONING SOCIAL ROLE/EVERYDAY LIFE BODY IMAGE EATING DISTURBANCES TREATMENT BURDEN HEALTH PERCEPTIONS RESPIRATORY SYMPTOMS DIGESTIVE SYMPTOMS WEIGHT	42 ± 25 (0–92) 33 ± 18 (0–75) 67 ± 18 (17–100) 56 ± 18 (22–100) 58 ± 26 (0–100) 67 ± 25 (22–100) 78 ± 24 (11–100) 56 ± 18 (11–100) 33 ± 22 (0–100) 39 ± 20 (0–78) 89 ± 20 (22–100) 67 ± 38 (0–100)	58 ± 26 (0–100) 50 ± 19 (8–100) 73 ± 19 (20–100) 56 ± 18 (11–100) 67 ± 24 (8–100) 67 ± 23 (11–100) 100 ± 20 (11–100) 56 ± 17 (11–100) 44 ± 21 (0–89) 56 ± 20 (0–89) 78 ± 18 (33–100) 100 ± 32 (0–100)	*p* ≤ 0.001 *p* ≤ 0.001 *p* ≤ 0.001 ns ns ns *p* ≤ 0.001 *p* ≤ 0.01 *p* ≤ 0.001 *p* ≤ 0.001 ns *p* ≤ 0.01
BMI (kg/m^2^) ^e^	19.8 ± 3.7 (12.8–33.7)	20.0 ± 3.7 (13.1–34.2)	*p* ≤ 0.01

^a^ All patients (*n* = 96). ^b^ CRP (*n* = 64) und Leukocyte count (*n* = 60) available on Day0 and Day28. ^c^ In-hospital lung function (*n* = 36) available on Day0 and Day28. ^d^ CFQ-R (*n* = 91) available on Day0 and Day28. ^e^ BMI (*n* = 67) available on Day0 and Day28.

**Table 4 antibiotics-12-00734-t004:** Outcome day 28. Inpatient and outpatient analysis shown for patients with complete data on days 0 and 28.

Follow up Day 28	Outpatients Median ± SD (Min–Max)	Inpatients Median ± SD (Min–Max)	*p*-Value
Home spirometry FEV1 (L/s) ^a^	1.9 ± 0.9 (0.9–4.8)	1.2 ± 0.8 (0.4–3.8)	*p* ≤ 0.01
PEx symptom score ^a^	1 ± 1 (0–5)	2± 2 (0–6)	*p* ≤ 0.05
Laboratory parameters ^b^: CRP (mg/L) Leukocyte count (/nL)	7.1 ± 12.1 (0.3–45.0) 10.4 ± 3.9 (2.4–18.6)	15.6 ± 33.2 (0.3–195.3) 13.7 ± 5.7 (2.8–25.5)	ns *p* ≤ 0.05
CFQ-R domains (%) ^c^ PHYSICAL FUNCTIONING VITALITY EMOTIONAL FUNCTIONING SOCIAL ROLE/EVERYDAY LIFE BODY IMAGE EATING DISTURBANCES TREATMENT BURDEN HEALTH PERCEPTIONS RESPIRATORY SYMPTOMS DIGESTIVE SYMPTOMS WEIGHT	67 ± 23 (13–100) 50 ± 20 (8–100) 73 ± 18 (20–100) 61 ± 19 (11–100) 67 ± 19 (17–100) 67 ± 22 (22–100) 100 ± 13 (44–100) 56 ± 14 (33–89) 44 ± 19 (11–78) 56 ± 19 (11–83) 89 ± 16 (44–100) 100 ± 27 (0–100)	46 ± 26 (0–92) 50 ± 19 (8–83) 67 ± 19 (27–100) 56 ± 17 (22–94) 53 ± 25 (8–100) 56 ± 23 (11–100) 78 ± 22 (11–100) 56 ± 19 (11–100) 44 ± 22 (0–89) 56 ± 22 (0–89) 78 ± 19 (33–100) 67 ± 37 (0–100)	*p* ≤ 0.001 ns ns ns *p* ≤ 0.01 *p* ≤ 0.05 *p* ≤ 0.01 ns ns ns *p* ≤ 0.05 *p* ≤ 0.05
BMI (kg/m^2^) ^d^	20.6 ± 3.8 (17.8–16.4)	19.6 ± 3.3 (13.1–28.6)	*p* ≤ 0.05

^a^ Outpatients *n* = 41, inpatients *n* = 54. ^b^ Outpatients *n* = 29, inpatients *n* = 40. ^c^ Outpatients *n* = 40, inpatients *n* = 51. ^d^ Outpatients *n* = 30, inpatients *n* = 38.

**Table 5 antibiotics-12-00734-t005:** Response to antibiotic therapy. Analysis shown for patients with complete data on days 0 and 28 ^a^.

	Responder (*n* = 57)	Non-Responder (*n* = 39)	*p*-Value ^b^
Female Age (years) CF-related diabetes Exocrine pancreatic insufficiency Hospitalizations, past 2 years Exacerbations, past 2 years ΔFEV1, past 2 years ^d^	28 (49.1%) 28.1 ± 11.4 (8.8–57.9) 19 (33.3%) 52 (91.2) 2 ± 3 (0–12) 4 ± 4 (0–13) −6.3 ± 14.0 (−58.5–22.7)	22 (56.4%) 32.1 ± 11.7 (16.2–57.0) 19 (48.7%) 37 (94.9) 4 ± 4 (0–17) 5 ± 4 (0–18) 7.0 ± 15.6 (−56.5–28.9)	ns ^c^ ns ns ns ns ns ns
Airway colonization: *Pseudomonas aeruginosa* *Staphylococcus aureus* Fungi	34 (59.6) 26 (45.6) 47 (82.5)	26 (66.7) 15 (38.5) 26 (72.2)	ns ns ns
Modified Fuchs score, day 0 BMI Day 0 (kg/m^2^) Home spirometry FEV1, day 0 (L/s) In-hospital FEV1, day 0 (L/s) ^e^ CRP, day 0 (mg/L) ^f^ Leukocyte count, day 0 (/nL) ^g^	5 ± 1 (3–6) 19.8 ± 3.6 (12.2–33.5) 1.2 ± 0.8 (0.4–3.9) 1.3 ± 0.8 (0.5–3.4) 26.5 ± 34.7 (0.5–177.5) 12.6 ± 5.3 (4.1–29.3)	4 ± 1 (3–6) 20.0 ± 3.8 (14.4–33.7) 1.3 ± 0.7 (0.4–3.2) 1.6 ± 0.9 (0.5–3.7) 14.9 ± 24.7 (0.3–100.4) 12.9 ± 5.0 (4.8–25.0)	*p* < 0.05 ns ns ns *p* < 0.05 ns
Outpatient treatment Duration of therapy Steroid therapy	25 (43.9) 14 ± 6 (10–28) 25 (43.9)	17 (43.6) 14 ± 5 (10–28) 13 (33.3)	ns ns ns
BMI-change, day 0–28 (kg/m²) ^h^ CRP-change, day 0–28 (mg/dL) ^f^ Leukocyte count change, day 0–28 (/nL) ^f^ Modified Fuchs score change, day 0–28	0.3 ± 1.0 (−0.7–4.1) −18.9 ± 38.2 (−174.1–15.8) −1.4 ± 3.2 (−7.6–6.5) −4 ± 2 (−6–2)	0.0 ± 0.8 (−3.0–1.4) 0.0 ± 25.6 (−42.8–101.2) 0.4 ± 4.0 (−9.0–10.0) −2 ± 2 (−5–1)	ns *p* < 0.001 ns *p* < 0.001

^a^ Data are displayed in the form of median ± SD (min–max) or number (%). ^b^
*p*-values refer to significances between inpatients and outpatients. ^c^ ns = not significant ^d^
*n* = 79 (responder *n* = 46; non-responder *n* = 33). ^e^
*n* = 59 (responder *n* = 35; non-responder *n* = 24). ^f^
*n* = 64 (responder *n* = 34; non-responder *n* = 30). ^g^
*n* = 60 (responder *n* = 32; non-responder *n* = 28). ^h^
*n* = 67 (responder *n* = 39; non-responder *n* = 28).

## Data Availability

Not applicable.

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
