# Peer review of "Antibiotic Therapy for Pulmonary Exacerbations in Cystic Fibrosis—A Single-Centre Prospective Observational Study"

_antibiotics, 2023, doi:10.3390/antibiotics12040734_

Round 1

Author Response

Summary section:

  1. Eliminate acronyms or abbreviations

Thank you for the comment. Acronyms and abbreviations have been eliminated so far, only FEV1 has been left as acronym for better readability, and as it is a widely used MESH term.

  1. The last sentence is unnecessary in this section.

We removed this sentence.

  1. Keywords must be MESH terms

In the MESH thesaurus, the word exacerbation is not listed primarily, however flare up (as listed) is not a common term for lung diseases.

 Introduction section:

  1. Homogenize throughout the text if the term bronchopulmonary exacerbation or pulmonary

exacerbation corresponds.

Thank you for this note. We homogenized the term into bronchopulmonary exacerbation.

  1. The objectives mentioned in this section do not correspond to the objectives mentioned in

the abstract.

We adapted the objectives in both sections.

  1. The information written as text of results is redundant with what is already mentioned in

table 1.

Thank you for this advice. The result section has been restructured and shortened, and we have chosen to dispense with the double data in the text and in tables and figures. The exact data can now only be found in the tables and figures. Additionally, we removed the figures presenting antibiotic treatment and microbiology results, since these provide only little additional information.

  1. The numbering of the tables is wrong.

We corrected and updated the numbering of all tables and figures.

Discussion section:

8.The first 02 paragraphs of this section are data that can go in the introduction section.

The first paragraph has been included in the discussion section, the second been removed due to redundancy.

  1. The discussion does not bear much relation to the objectives of the study. It needs to be

restructured.

We restructured and rewrote the discussion section.

Methodology section:

  1. Why was the range of over 6 years and under 75 years of age chosen as inclusion criteria?

The range was chosen, because spirometry was an important outcome parameter in our study and children under 6 years are not capable to perform reproducible spirometry. No standardized reference data for spirometry exist for this age group. The age limit of 75 years was a requirement of the ethics committee.

  1. The exclusion criteria are not a negation of the inclusion criteria.

We completed the missing criteria in both the in- and exclusion criteria part.

  1. It is not specified why the sample includes both children and adults nor how many

correspond to each age category

We are a mixed centre with children and adults with cystic fibrosis. Therefore, we included all patients who received antibiotic therapy for a bronchopulmonary exacerbation in our centre during the relevant period. The EU paediatric regulation of 2007 had the intention to enable more clinical trials in children, so we also attempted to include children in our study. The number of patients < 18 years is displayed in table 1.

Reviewer 2 Report

The current study ‘Antibiotic Therapy of Pulmonary Exacerbations in Cystic Fibrosis-A Single-Centre Prospective Observational Study’ evaluates Broncho pulmonary exacerbations (PEx) outcome parameters over 28 days in 96 pediatric and adult People with cystic fibrosis (pwCF) started on oral and/or intravenous antibiotic therapy after clinician diagnosis of PEx. PEx worsens long-term health status, but inpatient treatment is also associated with a poorer quality of life. However, the authors need to address the following points listed below:

 The authors need to rewrite the abstract as it is not clear.CF in the abstract? A brief summary of your results must be precisely written in the abstract.

Provide references to the studies described in line 81 "several studies."??

The result section appears to be overwritten and lengthy. In figure 2 “Antibiotic treatment” in the intravenous antibiotic therapy what does the orange color indicates? Modify the legend as well. Table 1 is mentioned two times in the manuscript.

In the section “2.4.2 Modified Fuchs PEx score” It is difficult to understand where these percentages are mentioned because they do not correspond to the provided figure and table. 

   In the subsection “2.4.4 Assessment of response to therapy” line 285 “fifty-seven patients were classified as treatment responders” is not mentioned in Table 1.

Provide a reason, for the analysis process why only female patients were chosen (as written in table 1). Material and methods should be elaborative and explanatory.

Authors must re-write the discussion part to explain their findings clearly.

Author Response

The current study ‘Antibiotic Therapy of Pulmonary Exacerbations in Cystic Fibrosis-A Single-Centre Prospective Observational Study’ evaluates Broncho pulmonary exacerbations (PEx) outcome parameters over 28 days in 96 pediatric and adult People with cystic fibrosis (pwCF) started on oral and/or intravenous antibiotic therapy after clinician diagnosis of PEx. PEx worsens long-term health status, but inpatient treatment is also associated with a poorer quality of life. However, the authors need to address the following points listed below:

 The authors need to rewrite the abstract as it is not clear.CF in the abstract? A brief summary of your results must be precisely written in the abstract.

Thank you for your comment. The abstract has now been rewritten and all results have been listed.

Provide references to the studies described in line 81 "several studies."??

As two studies are cited in the following section, we changed the text from “several” to “two” studies.

The result section appears to be overwritten and lengthy. In figure 2 “Antibiotic treatment” in the intravenous antibiotic therapy what does the orange color indicates? Modify the legend as well. Table 1 is mentioned two times in the manuscript.

Thank you for this advice. The result section has been restructured and shortened, and we have chosen to dispense with the double data in the text and in tables and figures. The exact data can now only be found in the tables and figures.

In the section “2.4.2 Modified Fuchs PEx score” It is difficult to understand where these percentages are mentioned because they do not correspond to the provided figure and table.

We have chosen to remove the double data in the text and in tables and figures. The exact data can now only be found in the tables and figures.

   In the subsection “2.4.4 Assessment of response to therapy” line 285 “fifty-seven patients were classified as treatment responders” is not mentioned in Table 1.

A table with the corresponding data has been added, and the subsection has been rewritten.

Provide a reason, for the analysis process why only female patients were chosen (as written in table 1). Material and methods should be elaborative and explanatory.

Female as well as male patients were chosen, but only female patients were mentioned to show the proportion of female as well as male patients. Both proportions are now included in the table.

Authors must re-write the discussion part to explain their findings clearly.

The discussion part has been restructured and rewritten.

Reviewer 3 Report

Dear authors

Tables seem to be un-editable an have low quality. However, if needed the authors can improve them. 

the quality of figures is also low. 

overall, findings are valid and suitable. 

Author Response

Tables seem to be un-editable an have low quality. However, if needed the authors can improve them. 

the quality of figures is also low. 

overall, findings are valid and suitable. 

Thank you for your comment. We provided tables as word document and will provide alternative figures if necessary.

Round 2

Reviewer 1 Report

No more comments

Reviewer 2 Report

The authors have addressed all my comments.